# The Effects of Intraoperative Hypothermia on Postoperative Cognitive Function in the Rat Hippocampus and Its Possible Mechanisms

**DOI:** 10.3390/brainsci12010096

**Published:** 2022-01-12

**Authors:** Guangyan Xu, Tianjia Li, Yuguang Huang

**Affiliations:** Department of Anesthesiology, Peking Union Medical College Hospital, Chinese Academy of Medical Sciences and Peking Union Medical College, Beijing 100730, China; bonnie_xgy08@163.com

**Keywords:** intraoperative hypothermia, postoperative cognitive dysfunction, hippocampus, synaptic plasticity, synaptic plasticity-related protein

## Abstract

Intraoperative hypothermia is a common complication during operations and is associated with several adverse events. Postoperative cognitive dysfunction (POCD) and its adverse consequences have drawn increasing attention in recent years. There are currently no relevant studies investigating the correlation between intraoperative hypothermia and POCD. The aim of this study was to assess the effects of intraoperative hypothermia on postoperative cognitive function in rats undergoing exploratory laparotomies and to investigate the possible related mechanisms. We used the Y-maze and Morris Water Maze (MWM) tests to assess the rats’ postoperative spatial working memory, spatial learning, and memory. The morphological changes in hippocampal neurons were examined by haematoxylin-eosin (HE) staining and hippocampal synaptic plasticity-related protein expression. Activity-regulated cytoskeletal-associated protein (Arc), cyclic adenosine monophosphate-response element-binding protein (CREB), S133-phosphorylated CREB (p-CREB [S133]), α-amino-3-hydroxy-5-methyl-4-isoxazole propionic acid receptor 1 (AMPAR1), and S831-phosphorylated AMPAR1 (p-AMPAR1 [S831]) were evaluated by Western blotting. Our results suggest a correlation between intraoperative hypothermia and POCD in rats and that intraoperative hypothermia may lead to POCD regarding impairments in spatial working memory, spatial learning, and memory. POCD induced by intraoperative hypothermia might be due to hippocampal neurons damage and decreased expression of synaptic plasticity-related proteins Arc, p-CREB (S133), and p-AMPAR1 (S831).

## 1. Introduction

Intraoperative hypothermia (defined as a core temperature <36 °C) is a common complication during operations; up to 25–70% of patients experience intraoperative hypothermia [1,2,3]. Numerous studies have revealed that intraoperative hypothermia is closely associated with several adverse events, including increased surgical blood loss and transfusion, shivering, morbid cardiac events, surgical site infection, delayed wound healing, and prolonged hospitalisation [4,5,6]. Recently, we published two independent observational studies, which revealed a high incidence of intraoperative hypothermia in China and identified its potential risk factors [2,7]. Additionally, we developed and validated the effectiveness of a risk prediction model, the Predictors Score, to determine patients’ hypothermia risk [8].

Postoperative cognitive dysfunction (POCD), a recognised clinical phenomenon characterised by impaired learning and memory, and decreased attention, has attracted increasing attention in recent years [9,10]. As one of the most common postoperative complications, POCD seriously affects patients’ postoperative rehabilitation, prolongs hospital stays, decreases long-term quality of life, and increases mortality, resulting in significant individual and societal burdens [11,12]. Although many mechanisms, including apoptosis [13], neuroinflammation and oxidative stress [11,14,15], autophagy [16], and synaptic plasticity dysfunction [16,17] are involved in POCD, the exact molecular mechanism remains unknown.

The hippocampus, which is integral to the formation and retention of episodic and spatial memory, is a brain region that is responsible for cognitive functions, such as learning, memory, and spatial navigation [18,19]. Several studies have demonstrated that amygdala–hippocampus–prefrontal cortex neural network is widely involved in emotion regulatory and memory consolidation processes [20,21]. Moreover, the neural circuitry of fear acquisition and extinction includes the amygdala, hippocampus, ventromedial prefrontal cortex (vmPFC) and dorsolateral prefrontal cortex (dlPFC), which is well established in rodent and humans [22,23].

Synaptic plasticity, the ability of synapses to modulate their strength or efficacy of synaptic transmission, underlies learning, memory, and information processing in the brain [24]. Non-invasive brain stimulation (NIBS), an innovative set of technologies and techniques, was used to stimulate or alter synaptic plasticity, modify and enhance cognitive, behavioural, social, and emotional processes [25]. Recently, the effects of NIBS on attention and memory functions were widely examined [26,27].

Activity-regulated cytoskeletal-associated protein (Arc) is involved in multiple forms of synaptic plasticity, including long-term potentiation (LTP) and long-term depression (LTD) [28]. Previous findings have implicated that Arc promotes the hippocampal neuronal maturation critical for learning, memory storage, and memory consolidation [29,30].

Cyclic adenosine monophosphate-response element-binding protein (CREB) is widely involved in neuronal development, plasticity, survival and neuroprotection in the central nervous system, and participates in LTP formation [17,31]. Several studies have confirmed that CREB exhibits vital biological functions, including learning ability, memory formation and cognition regulation [31,32]. CREB activation is mainly regulated by its phosphorylation at S133. CREB phosphorylation has also been shown to mediate neuronal proliferation, plasticity, survival and differentiation [33,34].

Postsynaptic α-amino-3-hydroxy-5-methyl-4-isoxazole propionic acid receptors (AMPARs) are responsible for the majority of fast excitatory synaptic transmission in the brain. Previous studies have reported that AMPARs are assembled from four distinct subunits (GluA1-4), which are involved in several forms of neuronal plasticity [35]. AMPARs play an important role in synapse formation and stabilisation, and the regulation of functional AMPARs is the principal mechanism underlying synaptic plasticity [36]. It has also been suggested that proper synaptic localisation and dynamic trafficking of AMPARs play a crucial role in synaptic plasticity [37,38]. S831-phosphorylated AMPAR1 regulates AMPAR functions, which are necessary for synaptic plasticity, spatial learning, and memory functions [39]. Hence, Arc, CREB, and AMPAR1 are functional synaptic proteins related to synaptic plasticity.

Currently, there are few specific studies addressing the correlation between intraoperative hypothermia and POCD. Based on the above clinical and basic studies, we sought to further explore the relationship between intraoperative hypothermia and postoperative cognitive function in rodents. The purpose of this study was to explore the effects of intraoperative hypothermia on postoperative cognitive function in rats undergoing exploratory laparotomies and to explore the possible molecular mechanisms. 

## 2. Materials and Methods

### 2.1. Animals

Adult male Sprague Dawley rats (300–400 g, aged 10–12 weeks) were purchased from HFK Bioscience Co., Ltd., Beijing, China. All rats were housed in a specific pathogen-free environment (room temperature 23 ± 1 °C, standard 12 h light/dark cycle, two rats per cage) with food and water available ad libitum. All animal procedures were approved by the Ethical Committee for Animal Experimentation of the Peking Union Medical College Hospital (XHDW-2019-003) and conformed to the Guide for the Care and Use of Laboratory Animals by the National Institutes of Health. 

### 2.2. Protocols

The animals were allowed to acclimate to the environment for a minimum of 3 days prior to the beginning of exploratory laparotomies. Rats were randomly assigned to three groups as follows (*n* = 10 per group): the hypothermic group, the normothermic group and the naïve group. Postoperatively, the rats were allowed to recover for 5 days without any manipulation. The rats then underwent behavioural tests starting with the Y-maze spontaneous alteration test on day 6, followed by the Morris Water Maze (MWM) test. The MWM test was carried out for five successive days as follows: training on days 7–10 followed by the probe trial on day 11. Upon completing the behavioural tests, all rats were euthanised by intraperitoneal injection of pentobarbital sodium to collect blood samples and hippocampal tissues for biochemical and molecular studies. A schematic of the experimental protocol is summarised in Figure 1A.

### 2.3. Surgery

Rats were anaesthetised with 40 mg/kg sodium pentobarbital (Beijing Sunbiotech Co., Ltd., Beijing, China) administered intraperitoneally, followed by exploratory laparotomies. Briefly, the rats in the hypothermic and normothermic groups were shaved, and an abdominal median incision of approximately 3 cm was made to allow penetration of the peritoneal cavity. The operator then explored the small intestine, and 5–10 cm of the small intestine was exteriorised and left in air for at 60 min. The intestine was then placed inside the peritoneal cavity, and the wound was sutured with 4-0 non-absorbable sutures in three layers consisting of the peritoneal lining, abdominal muscles, and skin. Core body temperatures were measured every 5 min throughout the operation using a rectal temperature probe. The entire surgical procedures lasted for at least 60 min. The body temperatures in hypothermic rats were maintained at 33 ± 0.5 °C by spraying 75% alcohol onto the rat’s body, while body temperatures of rats in the normothermic group were maintained at 37 ± 0.5 °C with a heating pad during operation. After the wounds were sutured, the rats were immediately intraperitoneally injected with penicillin (60,000 U) to prevent postoperative infections, and the body temperatures of those in the hypothermic group were raised to 37 ± 0.5 °C using a heating pad. For the rats in the naïve group, no operation was performed. 

### 2.4. Behavioural Tests

According to the experimental protocol, the postoperative cognitive function of the rats was evaluated by the Y-maze and MWM tests from postoperative days 6–11. All the behavioural experiments were conducted each day, and the camera was tracked using the EthoVision tracking system (EthoVision XT9, Noldus). The MWM test is the gold standard for assessing cognitive function in rodents, especially for testing hippocampus-related spatial learning and memory [40,41]. The Y-maze spontaneous alteration test is a hippocampal-dependent spatial working memory test used to measure the willingness of rodents to explore new environments [42,43]. In our experiment, the MWM and Y-maze tests were performed as previously reported with minor modifications [41,43]. 

### 2.5. Y-Maze Test

The Y-maze apparatus, made of grey plastic, has three identical arms separated at 120° angles (arm dimensions: 50 × 10 × 40 cm). The apparatus was used to evaluate the spontaneous alteration performance of rats at 6 days postoperatively. A representative image of the Y-maze apparatus is shown in Figure 1B. Each rat was placed in the centre of the Y-maze and was free to explore the three different arms of the maze for 8 min. After each test, the apparatus was thoroughly cleaned with 75% ethanol. Arm entry was recorded when the rat placed all four paws within the arm. The sequence and the total number of arms entered were recorded using a digital camera. Spontaneous alteration behaviour was determined from successive consecutive entries to the three different arms on overlapping triads in which all arms were represented. For example, a sequence of entries to the three arms A, B, and C, ACBABACABA, would generate four “successful” spontaneous alterations: ACB, BAC, and CAB. The percentage of spontaneous alterations was determined using the following equation:Spontaneous alteration percentage = ((number of spontaneous alterations)/(total arm entries-2)) − 100%

### 2.6. MWM Test

The MWM consisted of a circular pool (diameter: 150 cm, height: 50 cm; black interior wall) filled with water (depth: 25 cm; temperature: 22 ± 1.0 °C) and a circular platform (diameter: 10 cm) for the rats to escape. The pool was divided into four equal-sized virtual quadrants (1, 2, 3, and 4), and the removable escape platform was placed 2 cm below the water’s surface in the first quadrant (Figure 1C). Furthermore, the water was made opaque by adding nontoxic black ink to prevent the rats from seeing the submerged platform. In each training trial, the rat was gently placed into the water facing the tank wall at the central point of the quadrant. Then, each rat was allowed to swim for 60 s to find the hidden platform and was subjected to three consecutive trials from quadrants 2–4 per day. When successful, the rats were allowed a 15 s rest period on the platform. If unsuccessful within the allotted time, the rat was guided to the platform and allowed to rest for 15 s on the platform. For all training trials, the average swimming speed and the mean time to reach the platform (escape latency) across three trials were noted as the daily result of learning ability for the rats, which were recorded and analysed by the computerised tracking system. 

On the 11th day, the rats were additionally tested for spatial learning and memory abilities by removing the underwater platform from the pool. The rat was gently placed in the water at the central point of the quadrant 3, opposite the platform location, and had 60 s to search for the platform’s original location. The time that each rat spent searching for the platform in the target quadrant and the number of platform crossings were recorded.

### 2.7. Haematoxylin-Eosin (HE) Staining

To assess the morphological changes in the rats’ hippocampal neurons, brain tissues were harvested after the rats were deeply anaesthetised with intraperitoneal pentobarbital sodium injections (40 mg/kg). The rats were transcardially perfused with sterile phosphate-buffered saline (PBS), followed by 4 °C paraformaldehyde (Sigma, St. Louis, MO, USA). Air bubbles were avoided throughout the perfusion. The rats’ brains were then dissected and post-fixed for at least 24 h in 4% paraformaldehyde at 4 °C. Subsequently, the tissue samples were dehydrated using graded alcohol steps, immersed in xylene, embedded in paraffin, sectioned along the coronal plane at 5 μm thickness (Leica 2000, Leica Microsystems, Wetzlar, Germany), and stained with HE. Images were examined under bright-field illumination using a fluorescence microscopy imaging system (Ti-S, Olympus FluoView Software, Olympus, Tokyo, Japan). The total number of injured neurons in each image was counted and analysed by Image-Pro Plus 6.0 Software.

### 2.8. Western Blot Analysis

After deep anaesthesia with pentobarbital sodium (40 mg/kg) and transcardial perfusion with sterile PBS, the brains were rapidly removed, and bilateral hippocampal tissues were dissected and flash-frozen in liquid nitrogen. Hippocampal tissues were homogenised in Tissue Protein Extraction Reagent (Thermo Fisher Scientific, Waltham, MA, USA) with a protease inhibitor cocktail (CWbio, Beijing, China) and phosphatase inhibitor cocktail (CWbio, Beijing, China). After centrifugation (12000× *g* for 15 min at 4 °C), the supernatants were collected and denatured in sodium dodecyl sulphate-polyacrylamide gel electrophoresis (SDS-PAGE) loading buffer (Solarbio, Beijing, China) for 10 min at 100 °C. Protein concentrations were determined using the Pierce BCA Protein Assay (Thermo Scientific, Rockford, IL, USA).

The protein samples were separated by SDS-PAGE and transferred to a polyvinylidene fluoride membrane (Millipore, Billerica, MA, USA). The membranes were blocked with 5% non-fat dry milk (Solarbio, Beijing, China) or 5% bovine serum albumin (Solarbio, Beijing, China) in Tris-buffered saline with 0.5% Tween 20 (TBST, Solarbio, Beijing, China) for 1 h at room temperature and incubated with primary antibodies overnight at 4 °C. The following primary antibodies were used: mouse anti-Arc (sc-17839, 1:500, Santa Cruz, CA, USA), rabbit anti-CREB (ab32515, 1:1000, Abcam, Cambridge, MA, USA), rabbit anti-S133-phosphorylated CREB (p-CREB [S133]; #9198, 1:1000, Cell Signaling Technology, Danvers, MA, USA), rabbit anti-AMPAR1 (ab183797, 1:1000, Abcam, Cambridge, MA, USA), rabbit anti-S831-phosphorylated AMPAR1 (p-AMPAR1 [S831]; ab109464, 1:1000, Abcam, Cambridge, MA, USA), and mouse anti-β-actin (cat No. 66009-1-lg, 1:5000; Proteintech Group, Chicago, IL, USA). 

The membranes were subsequently washed three times for 5 min each in TBST (Solarbio, Beijing, China) and incubated for 1 h at room temperature with the following secondary antibodies: HRP-conjugated affinipure goat anti-rabbit IgG (H+L; cat No. SA00001-2, 1:5000, Proteintech Group, Chicago, IL, USA), and HRP-conjugated affinipure goat anti-mouse IgG (H+L; cat no. SA00001-1, 1:5000, Proteintech Group, Chicago, IL, USA). The bands were visualised via a Tanon 5800 Luminescent Imaging Workstation (Tanon Science & Technology Co., Ltd., Shanghai, China) using High-sig ECL Western blotting Substrate (Solarbio, Beijing, China). The band intensity was measured using ImageJ software (National Institutes of Health, Bethesda, MD, USA). The ratio of each band/β-actin was considered as the expression level of the target protein.

### 2.9. Statistical Analysis

All data are presented as mean ± SEM. Statistical analysis was performed using GraphPad Prism 8 software (version 8.01, GraphPad Software, San Diego, CA, USA). The normality of the data distribution was determined by visual inspection of quantile-quantile (Q-Q) plot. Levene’s test was used to assess the homogeneity of variances. The rats’ escape latency and swimming speed in the MWM were analysed using two-way repeated measures analysis of variance (ANOVA) followed by Tukey’s post hoc multiple comparisons test. Other results were analysed using one-way ANOVA followed by Tukey’s post hoc test. In this study, *p* < 0.05 was considered statistically significant.

## 3. Results

### 3.1. Intraoperative Hypothermia Impaired Rats’ Spatial Working Memory

To assess the effect of intraoperative hypothermia on spatial working memory in the rat hippocampus, we used the Y-maze spontaneous alteration test, which utilises rats’ tendency to freely explore alternate maze arms in successive trials. Figure 2A shows the representative tracks of rats moving in the Y-maze in different groups. During the 8 min test session, there were no significant between-group differences in total arm entries (F_2,26_ = 0.7431, *p* = 0.4855; Figure 2B) and total distance moved (F_2,26_ = 2.378, *p* = 0.1126; Figure 2C) at 6 days postoperatively. However, a significant decrease in spontaneous alteration percentage was found in rats in the hypothermic group compared with the normothermic (*p* = 0.0392; Figure 2D) and naïve groups (*p* = 0.0168; Figure 2D), but there was no significant difference between the normothermic and naïve groups (*p* = 0.9223; Figure 2D). To summarise the above behavioural test results, the spatial working memory after exploratory laparotomy was compromised following intraoperative hypothermia; however, we could suppress the cognitive deficits by maintaining a relatively constant body temperature during the procedure.

### 3.2. Intraoperative Hypothermia Reduced Rats’ Postoperative Spatial Learning and Memory Ability

We conducted the MWM test to evaluate hippocampal-dependent spatial learning and memory of rats in each group. The representative swimming trajectories of rats from each group on the last day of the training trials and probe trials are shown in Figure 3A,B, respectively. During the consecutive 4-day training phase, we found that the escape latencies were obviously shortened in all groups (F_2,26_ = 4.669, *p* = 0.0185; Figure 3D). Specifically, the rats in the hypothermic group spent more time searching for the hidden platform than those in the naïve group on day 8 (*p* = 0.0415; Figure 3D). On day 11, in the probe test, the hypothermic rats spent less time in the target quadrant than the normothermic (*p* = 0.0031; Figure 3E) and naïve rats (*p* = 0.0041; Figure 3E), but there was no significant difference between the normothermic and naïve groups (*p* = 0.9997; Figure 3E). 

Moreover, the results revealed that the number of platform crossings was markedly reduced in the hypothermic group compared with that in the normothermic (*p* = 0.0343; Figure 3F) and naïve groups (*p* = 0.0002; Figure 3F). However, in the normothermic group, the number of crossings on the platform was not statistically different from that in the naïve group (*p* = 0.0893; Figure 3F). Nevertheless, there were no between-group differences in any training trials regarding the rats’ swimming speed (F_2,26_ = 1.926, *p* = 0.1660; Figure 3C). Taken together, these results suggest that intraoperative hypothermia could lead to spatial learning memory impairment in rats; however, maintaining the normal body temperature of rats during operation could ameliorate this cognitive decline. 

### 3.3. Intraoperative Hypothermia Increased Hippocampal Neuron Injuries in Postoperative Rats

In this study, we observed postoperative morphological changes of neurons in rats’ hippocampal dentate gyrus (DG) region by HE staining (Figure 4). We found the neurons in the naïve group were round or oval with rich, large, and lightly stained cytoplasm, and orderly arranged nuclei (Figure 4C). Similarly, we observed that the hippocampal neurons of rats in the normothermic group showed normal morphology, similar to the naïve group (*p* = 0.9983; Figure 4B,D). However, disorderly arranged nuclei, neuronal pyknosis and necrosis in the hippocampus were evident in the hypothermic group (Figure 4A, white arrows) compared to the naïve (*p* = 0.0222; Figure 4D) and normothermic (*p* = 0.0241; Figure 4D) groups. Our findings indicate that rats’ postoperative cognitive function may be related to the morphological changes of hippocampal neurons, and intraoperative hypothermia could significantly damage hippocampal nerve cells. This suggests that maintaining rats’ body temperature within the physiological range could alleviate hippocampal neuronal damage.

### 3.4. Intraoperative Hypothermia Suppressed the Expression of Synaptic Plasticity-Related Proteins in the Hippocampal Region of Rats

To further elucidate the underlying molecular mechanism, the expression levels of synaptic plasticity-related proteins such as Arc, CREB, and AMPAR1 in the hippocampi of rats were examined by Western blotting. We also determined CREB and AMPAR1 activities by measuring p-CREB (S133) and p-AMPAR1 (S831) levels in the hippocampal extracts of rats. The hypothermic group exhibited significantly lower levels of Arc protein expression than the normothermic (*p* = 0.0117; Figure 5D) and naïve groups (*p* = 0.0005; Figure 5D), although there was no significant difference between the normothermic and naïve groups (*p* = 0.1128; Figure 5D). Moreover, no remarkable between-group differences were found regarding CREB expression levels (F_2,9_ = 0.05574, *p* = 0.9641; Figure 5B) and AMPAR1 (F_2,9_ = 3.147, *p* = 0.0920; Figure 5E). 

However, after the band intensities of p-CREB (S133) were normalised to CREB, we found that hypothermic rats displayed significantly lower p-CREB (S133) expression compared to the normothermic (*p* = 0.0466; Figure 5C) and naïve groups (*p* = 0.0028; Figure 5C), whereas no significant difference was found between the normothermic and naïve groups (*p* = 0.0819; Figure 5C). 

Additionally, we performed a quantitative analysis of the expression levels of p-AMPAR1 (S831) normalised to AMPAR1 band intensities and showed that the hypothermic group presented decreased levels of p-AMPAR1 (S831) compared to the normothermic (*p* = 0.0470; Figure 5F) and naïve groups (*p* = 0.0081; Figure 5F). However, there was no significant difference between the normothermic and naïve groups (*p* = 0.5133; Figure 5F). 

## 4. Discussion

This study was designed to explore the effects of intraoperative hypothermia on postoperative cognitive function in rats undergoing exploratory laparotomies and to investigate the related molecular mechanisms. In our study, hypothermic rats showed impaired spatial working memory (Figure 2), as demonstrated by the reduced spontaneous alteration percentage evaluated by the Y-maze test. Notably, when the rats’ temperature was maintained within a normal range (37 ± 0.5 °C) during surgical operation, the spontaneous alteration percentage of rats in the normothermic group was statistically indistinguishable from that of naïve rats (Figure 2), suggesting that maintaining core temperature in a normal range could dramatically alleviate the detrimental effects of intraoperative hypothermia on rats’ spatial working memory. Similarly, in the MWM test, rats in the hypothermic group exhibited markedly decreased spatial learning and memory ability, as evidenced by significant reductions in time spent in target quadrant and number of platform crossings, comparable to the normothermic and naïve groups (Figure 3). However, no significant differences were observed in time spent in target quadrant and number of platform crossings between the normothermic and naïve groups, indicating that intraoperative hypothermia seriously affects spatial learning and memory in rats.

To further examine the possible mechanisms behind these behavioural phenomena, we performed HE staining to examine neuronal integrity and orderliness in the rats’ hippocampi. Our histopathological results revealed significant damage in the hippocampi of hypothermic rats (Figure 4A,D), which showed disorderly nuclei, neuronal pyknosis and necrosis. However, the hippocampal tissues taken from rats in the normothermic (Figure 4B,D) and naïve groups (Figure 4C,D) were similar, showing normal pyramidal nerve cell morphology, confirming that intraoperative hypothermia could induce POCD by damaging hippocampal neurons, and that maintaining normal body temperatures could alleviate POCD and hippocampal neuronal damage.

Multiple studies have established that synaptic plasticity plays a critical role in learning and memory by governing the number and strength of connections within a cell assembly [44,45,46]. Although various pathological factors have been demonstrated to be responsible for POCD, evidence from experimental studies revealed that synaptic dysfunction in the central nervous system, particularly in the hippocampus, may be involved in POCD’s progression [47,48]. Furthermore, synaptic dysfunction is associated with the abnormal expression of proteins required for synaptic function [49,50]. Therefore, we measured the expression levels of synaptic plasticity-associated proteins, including Arc, CREB, p-CREB (S133), AMPAR1, and p-AMPAR1 (S831) by Western blotting to further explore the underlying molecular mechanism.

We found that the expression of hippocampal synaptic plasticity-related proteins is consistent with rats’ behavioural results and histopathological outcomes. We observed a dramatic decrease in Arc levels in the hippocampi of hypothermic rats compared to normothermic and naïve rats (Figure 5D), whereas no differences were observed between the normothermic and naïve rats. It is well recognised that Arc plays a vital role in multiple forms of learning and memory by regulating synaptic plasticity [29,51]. Specifically, Arc was found to be strongly related to synaptic plasticity processes dependent on protein synthesis [52]. Moreover, Plath et al. suggested that Arc-knockout mice show severe long-term memory impairment [29]. Thus, the lower expression of Arc in the hypothermic group demonstrated the impairment of synaptic plasticity, learning, and memory in rats, consistent with the outcomes of the Y-maze spontaneous alteration and MWM tests. 

Subsequently, we measured the levels of CREB and phosphorylation of CREB at the Ser133 site in the rat hippocampus among the three groups and found significant p-CREB (S133) expression inhibition in hypothermic rats compared to normothermic and naïve rats (Figure 5C). However, no significant difference was found between normothermic and naïve rats. In addition, no significant between-group difference was found regarding the level of CREB expression (Figure 5B). Several studies have illustrated that CREB mobilisation is dependent on phosphorylation at the Ser133 site, which is widely involved in many forms of neuronal activity: synaptic plasticity, learning, and memory [53,54]. Lv et al. identified that Arc is the critical downstream molecule of the extracellular signal-related kinase (ERK)-CREB pathway [55]. Briefly, the above results show that a decrease in p-CREB (S133) is associated with intraoperative hypothermia-induced POCD, and lower levels of Arc may be caused by the reduction of p-CREB (S133).

Previous studies have reported that Arc has a close, bidirectional relationship with postsynaptic glutamate neurotransmission because it is stimulated by numerous glutamatergic receptor mechanisms and regulates AMPAR internalisation, trafficking, and activity-dependent scaling [56,57]. At excitatory synapses in the central nervous system, AMPARs are responsible for most synaptic transmission [51]. The effects on AMPAR trafficking are likely associated with Arc’s ability to mediate different forms of synaptic plasticity, such as LTP, LTD, and synaptic scaling, each of which is essential for normal cognitive function [52,58]. Therefore, AMPARs are critical downstream molecules of Arc.

In this study, to further explore Arc function, we detected AMPAR1 and p-AMPAR1 (S831) in the hippocampi of rats. We found that AMPAR1 protein expression differed slightly between the groups (Figure 5E). In addition, p-AMPAR1 (S831) was decreased in rat hippocampi after intraoperative hypothermia compared with the normothermic and naïve groups, whereas no differences were found between normothermic and naïve rats (Figure 5F). Overall, our results show that intraoperative hypothermia can significantly inhibit hippocampal synaptic plasticity-related proteins, including Arc, p-CREB (S133), and p-AMPAR1 (S831), indicating that intraoperative hypothermia impairs the postoperative cognitive function of rats by reducing the levels of hippocampal synaptic plasticity-related proteins.

To the best of our knowledge, there are few relevant studies exploring the effects of intraoperative hypothermia on postoperative cognitive function in rats undergoing exploratory laparotomies and investigating the related mechanisms. In our research, we found a correlation between intraoperative hypothermia and POCD in rats. The underlying mechanism may associate with hippocampal neuron damage and downregulation of synaptic plasticity-associated proteins Arc, p-CREB (S133), and p-AMPAR1 (S831). When interpreting these results, it is important to consider some limitations. First, POCD, as a commonly used clinical term, has some limitations when used to describe impaired cognitive function in rodents after operations. In clinical research studies, the Montreal Cognitive Assessment (MoCA) and the Mini-Mental State Examination (MMSE) are the most commonly used neuropsychological rating scales for the diagnosis of POCD. However, such behavioural tests as the Y-maze and MWM tests are two commonly used evaluation methods to examine spatial cognitive functions of rodents in basic research. It is debatable whether the two types of evaluation methods present concordance both in clinical research and basic research for a valid diagnosis of POCD. In addition, we are only presenting a preliminary exploration of the mechanisms underlying intraoperative hypothermia effect on POCD; the precise mechanism requires further investigation.

## 5. Conclusions

POCD has been shown to be associated with impaired concentration, memory, and learning. Intraoperative hypothermia and its adverse consequences have drawn increasing attention in recent years. We found a correlation between intraoperative hypothermia and POCD in rats. Specifically, our study indicated that intraoperative hypothermia could cause POCD, affecting rats’ spatial working memory, spatial learning, and memory. POCD induced by intraoperative hypothermia might be due to hippocampal neuron damage and the reduced expression of synaptic plasticity-related proteins, such as Arc, p-CREB (S133), and p-AMPAR1 (S831). This study was a preliminary exploration of the mechanisms underlying intraoperative hypothermia effect on POCD; the precise mechanism requires further investigation.

## Figures and Tables

**Figure 1 brainsci-12-00096-f001:**
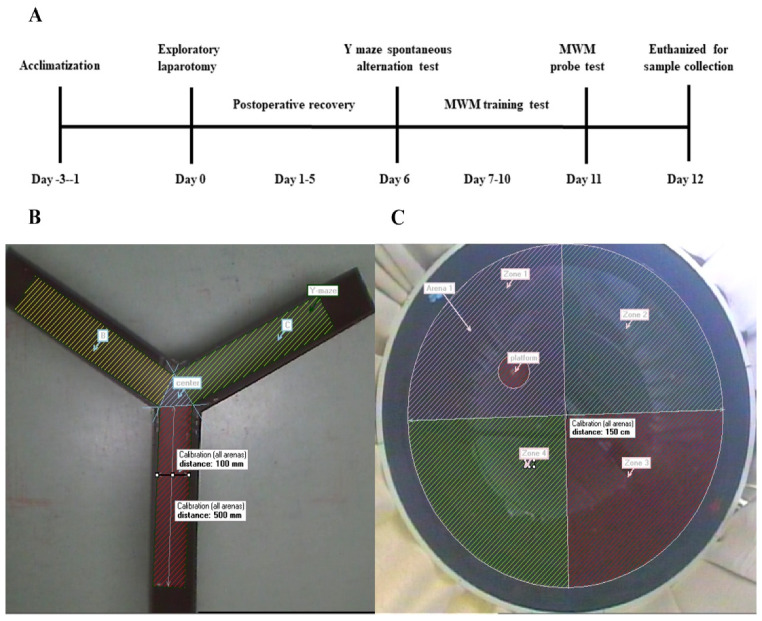
Study design to evaluate the effects of intraoperative hypothermia on postoperative cognitive function in rats. (**A**) Schematic timeline of the experimental procedure; (**B**) representative image of Y-maze apparatus; (**C**) representative graph of the MWM apparatus.

**Figure 2 brainsci-12-00096-f002:**
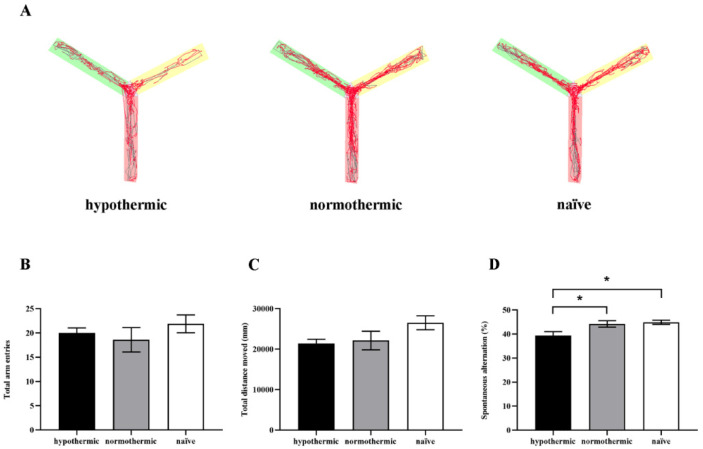
The effects of intraoperative hypothermia on the spatial working memory of rats measured by the Y-maze spontaneous alternation. (**A**) Representative mobile trajectories of rats among different groups; (**B**) the number of total arm entries of rats from different groups; (**C**) the total distance moved during 8 min free exploration in the Y-maze; (**D**) the spontaneous alternation percentage in different groups. Data are expressed as mean ± SEM. * *p* < 0.05, *n* = 9–10.

**Figure 3 brainsci-12-00096-f003:**
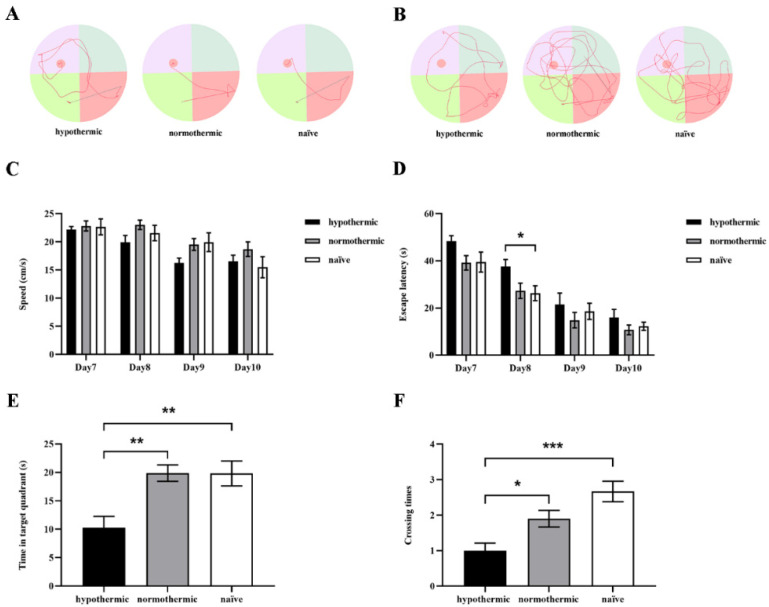
The effects of intraoperative hypothermia on spatial learning and memory of rats were determined by the MWM. (**A**) The representative swimming trajectories of rats from each group on the last day of training trials; (**B**) the representative swimming paths of rats from each group in the probe test; (**C**) the average swimming speeds of rats in different groups on successive MWM training days; (**D**) the average escape latencies to find the hidden platform across three trials on four consecutive training days; (**E**) the time spent in the target quadrant in the probe test; (**F**) the number of platform crossings during the probe test. Data are expressed as mean ± SEM. * *p* < 0.05, ** *p* < 0.01 and *** *p* < 0.001, *n* = 9–10.

**Figure 4 brainsci-12-00096-f004:**
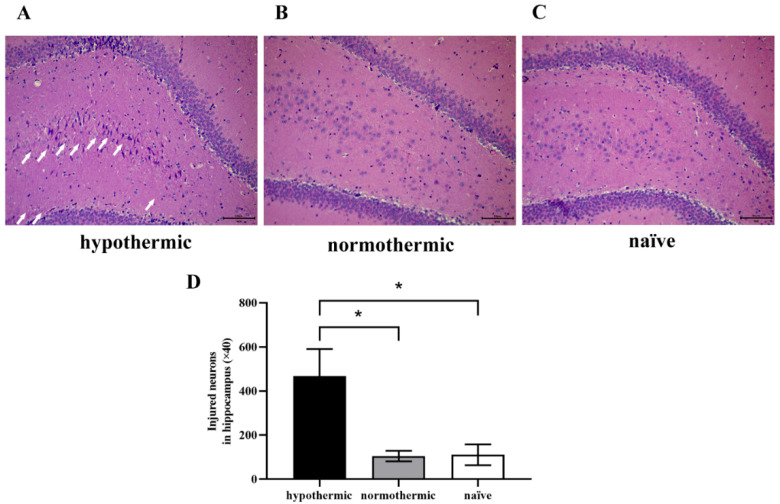
Morphological changes of hippocampal neurons in rats among groups were observed under bright-field illumination using a fluorescence microscopy imaging system. (**A**–**C**) Representative images of HE staining in hippocampal slices; (**D**) quantitative analysis of injured neurons with neuronal pyknosis and necrosis in the hippocampal DG region. The white arrow indicates a disorderly nucleus, neuronal pyknosis and necrosis. Magnification: ×40; scale bar: 50 µm. Data are expressed as mean ± SEM. * *p* < 0.05, *n* = 4.

**Figure 5 brainsci-12-00096-f005:**
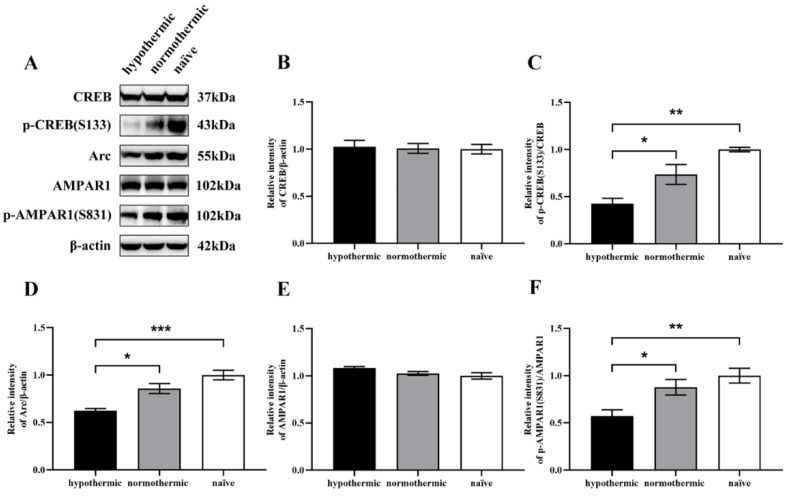
Expression levels of hippocampal synaptic plasticity-related proteins in rats and the effects of intraoperative hypothermia. (**A**) The visualisation of protein bands of CREB, p-CREB (S133), Arc, AMPAR1, p-AMPAR1 (S831), and β-actin; (**B**) quantitative analysis of CREB expression levels normalised to β-actin band intensities; (**C**) quantitative analysis of p-CREB (S133) expression levels normalised to CREB band intensities; (**D**) quantitative analysis of Arc expression levels normalised to β-actin band intensities; (**E**) quantitative analysis of AMPAR1 expression levels normalised to β-actin band intensities; (**F**) quantitative analysis of p-AMPAR1 (S831) expression levels normalised to AMPAR1 band intensities. Data are expressed as mean ± SEM. * *p* < 0.05, ** *p* < 0.01 and *** *p* < 0.001, *n* = 3–4.

## Data Availability

All data included in this study are available upon request from the corresponding authors.

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
