# Peer review of "The Effects of Intraoperative Hypothermia on Postoperative Cognitive Function in the Rat Hippocampus and Its Possible Mechanisms"

_brainsci, 2022, doi:10.3390/brainsci12010096_

Round 1

Reviewer 1 Report

In the present study entitled ‘The effects of intraoperative hypothermia on postoperative cognitive function in the rat hippocampus and its possible mechanisms’, by Xu and colleagues, authors addressed the possible effects of intraoperative hypothermia on postoperative cognitive function (POCD), and related mechanisms, in rats. For this purpose, the experiment was conducted in thirty rats, that were divided into three experimental groups: the hypothermic group, the normothermic group, and the naïve group. The rats' behavior and pathological condition were detected using Y-maze, Morris Water Maze (MWM), and hematoxylin staining. Results showed a correlation between intraoperative hypothermia and postoperative cognitive function in rats and that intraoperative hypothermia may lead to POCD regarding impairments in spatial working memory, spatial learning, and memory.

In general, I think the idea of this article is really interesting and the authors’ fascinating observations on this timely topic may be of interest to the readers of Brain Sciences. However, some comments, as well as some crucial citations that should be included to support the authors’ argumentation, need to be addressed to improve the article, its adequacy, and its readability prior in the present form. My overall judgment is that authors have carefully considered my suggestions below, in particular regarding the Introduction and the Discussion sections.

Comments

  • Page 2-3, Introduction: The authors decided to include only animal models to discuss the importance of hippocampal neuronal maturation in learning, memory storage and memory consolidation. Still, I think that the Introduction of this study is not enough extensive and it does not seem to consider, in most cases, all the available studies in the literature that have acknowledged hippocampal contribution to cognition in humans as well. Hence, I suggest that a general overview on underlying mechanisms associated with neural activity in brain areas required for associative learning and memory processing in humans too might be crucially useful. For these reasons, I would suggest some references that would be crucial and methodologically fit with the present manuscript: for example, Borgomaneri and colleagues’ study (2020, Current Biology - https://doi.org/10.1016/j.cub.2020.06.091) showed that the inhibition of the dorsolateral prefrontal cortex (DLPFC, that directly interacts with the hippocampus and amygdala) after memory reactivation disrupts physiological responding to learned fear, highlighting the role of this area in the neural network that mediates the reconsolidation of fear memories in humans. Similarly, I also suggest mentioning a recent review (Borgomaneri et al., 2021, Neuroscience and Biobehavioral Reviews - https://doi.org/10.1016/j.neubiorev.2021.04.036) on the ability of non-invasive brain simulation (NIBS) to interfere with activity of neural circuits (i.e., amygdala-mPFC-hippocampus) involved in the acquisition and consolidation of memories. Moreover, Borgomaneri and colleagues’ study (2021, Journal of Affective Disorders - https://doi.org/10.1016/j.jad.2021.02.076) illustrated the therapeutic potential of NIBS as a valid alternative in the treatment of abnormally persistent memories that characterized those patients with anxiety disorders that do not respond to psychotherapy and/or drug treatments, usually patients with an alteration of the amygdala-mPFC-hippocampus neural network. Finally, in a recent study that involved patients with brain lesions, Battaglia and colleagues (2020, The Journal of Neuroscience - https://doi.org/10.1523/JNEUROSCI.0304-20.2020) provided evidence that the ventromedial prefrontal cortex (vmPFC) is involved in the acquisition of fear conditioning (i.e., learning). Also, the role of vmPFC in processing safety-threat information and their relative value, and how this region is fundamental for the evaluation and representation of stimulus-outcome’s value needed to produce sustained physiological responses, it has also been discussed in a recent review on vmPFC subregional contributions (Battaglia et al., 2021, Molecular Psychiatry -https://doi.org/10.1038/s41380-021-01326-4). Moreover, if the authors consider it appropriate, they can also see Brunoni and Vanderhasselt’ review (2014, Brain and cognition) on memory improvement using non-invasive brain stimulation, or the reviews by Takeuchi and colleagues (2013, Philosophical transactions of the Royal Society of London. Series B, Biological sciences) and by Bailey and colleagues (2015, Cold Spring Harbor Perspectives in Biology) about synaptic plasticity in mediating memory formation.
  • Page 1, lines 48-50: Authors stated that ‘Synaptic plasticity, the ability of synapses to modulate their strength or efficacy of synaptic transmission, underlies learning, memory, and information processing in the brain’. In this regard, I recommend studies that could provide interesting insight on the use of non-invasive brain stimulation (NIBS), an innovative set of technologies and techniques used to stimulate or alter synaptic plasticity and, therefore, to modify and enhance cognitive, behavioral, social, and emotional processes: in a recent study, Borgomaneri and colleagues (2021, Brain Sciences - https://doi.org/10.3390/brainsci11091203) provided interesting insights on the use of single-pulse TMS to investigate the time course of the motor system readiness to relevant arousing stimuli (i.e., happy and fearful faces), addressing how differences in the experience of aversive feelings modulate corticospinal excitability, hence, the preparation of adaptive motor responses required for the execution of appropriate behaviors. Similarly, Borgomaneri and colleagues (2020, Cortex - https://doi.org/10.1016/j.cortex.2020.09.002) outlined how NIBS techniques can be used to manipulate the ability to manipulate prepotent ongoing motor actions in healthy individuals. Accordingly, in another work, Battaglia and colleagues (2021, Behaviour Research and Therapy - https://doi.org/10.1016/j.brat.2021.103963) described typical dysfunctional behaviors, such as deficit in action control and motor inhibition, that are associated with psychopathological and psychiatric conditions, which are characterized by impulsivity problems (which can be intensified in the presence of emotional stimuli). I believe that adding this series of study will dramatically improve the theoretical background of the present article and its argumentation. Moreover, if the authors deem it appropriate, they can also see a review on the implementation of NIBS by Sandrini and Cohen, (2013, Handbook of clinical neurology), or the review by Hara and colleagues (2021, Diagnostics). Also, the studies by Lynch and colleagues (2019, Cerebral Cortex), Roschchupkina and colleagues (2020, Brain stimulation) and the review by Vosskuhl and colleagues (2018, Frontiers in human neuroscience) that examined the effects of NIBS on attention and memory functions may be of interest.
  • Regarding the Results: In my opinion, the 'Results’ section is well organized, but it seems to state the statistical significance of findings in an excessively broad way. Thus, I believe that this section would benefit from a more detailed and precise rewriting, in order to ensure an in-depth understanding of the findings.
  • Page 10-11, Discussion: I suggest moving the opening paragraphs in the Introduction section; indeed, here authors outline the effectiveness of a risk model and describe hypothermia’s clinical parameters, information that I believe should be included in the background of the study.
  • Regarding the Conclusion: In my opinion, this section is too small and contains too many broad statements to adequately convey what the writers believe is the take-home message. To begin with, I would like to read here a brief summarization about evidence of postoperative cognitive dysfunction (POCD) and a brief description of the associated impaired cognitive domains (e.g., attention, learning and memory); therefore, I think that this section would benefit from more precise as well as in-depth considerations.
  • In according to the previous comment, I would ask the authors to include a ‘Limitations and future directions’ section before the end of the manuscript, in which authors can describe in detail and report all the technical issues brought to the surface.
  • Regarding the Figures and the Tables: please provide a short explanatory title for each figure and table in the main text. Also, I suggest providing higher-resolution images of Y-maze and of the MWM apparatus in Figure 1, to help readers better comprehend them.

Reviewer 2 Report

The current manuscript is an interesting study which investigated the effect of intraoperative hypothermia on postoperative cognitive function by focusing on the hippocampus. They found that there is a significant positive correlation between the intraoperative hypothermia and the impaired postoperative cognitive function and the mechanistically approach showed involvement of hippocampal neurons damage and decreased expression of synaptic plasticity-related proteins Arc, p-CREB (S133), and p-AMPAR1 (S831). This study includes useful and important information, however, there are several issues which are needed to be addressed:

  1. It is not clear which sex of rats was used in this study.
  2. Authors used the same does of pentobarbital (40 mg/kg) for both surgical anesthesia and sac the animals to collect the tissue?
  3. For cutting the paraffin sections, it is needed to prepare the paraffin blocks of tissue which includes several steps of dehydration, however, authors put the brains in the 30% sucrose which is used for cryo sectioning.
  4. It seems that authors used some of the animals for morphological analysis, and 3-4 of them for molecular analysis which is very low number of animals per group and will be resulted in a low power.
  5. How is the fraction between the cut sections? How many sections were cut?
  6. How the normality of data distribution was checked? How the homogeneity of variances are checked?
  7. In the result section F2,26 = 5.151, p = 0.0392; Figure 2D) and naïve groups (F2,26 = 5.151, p = 0.0168; Figure 2D), they have two different p values with the same F?

The same here: (F2,26 = 8.772, p = 0.0031; 249 Figure 3E) and naïve rats (F2,26 = 8.772, p = 0.0041; Figure 3E), but there was no significant difference between the normothermic and naïve groups (F2,26 = 8.772, p = 0.9997)

  1. Regarding the behavioral results, it is not clear what the independent factors are? how is the interaction between independent factors and how is the main effect of independent factors?
  2. What does this mean? (F216,57.62 = 58.10, p < 0.0001; Figure 3D).
  • Authors says Intraoperative hypothermia increased hippocampal neuron injuries in postoperative rats, however, the results are about morphological changes and not the number , and what is the meaning of neuronal injury?
  • What is the p value for the neuronal morphology difference between the groups?
  • Which part of hippocampus is investigated as pyramidal neurons have different morphology than granular neurons.
  • Which microscopic lens was used to do morphological quantification of neurons? Which method is used for the morphological quantification?

Reviewer 3 Report

Thanks for the work. The following major corrections are required.
1) There are more than 50 references in the Introduction, this density makes the Intro information very low. It is expected that there are 15-20 references on average, and these references are current, high-impact and up-to-date reviews. If the literature is to be discussed in between, this is supported by 1-2 research articles. The intro should be re-written.
2) Unfortunately, all of the figures are problematic. The resolution is very poor. Nothing is clear from the pictures shown. For example, a figure has inscriptions but is never read. They should be zoomed in - paying attention to resolution.
3) Discussion should be written in an appropriate order from the beginning.
First, the most important data will be given and discussed with the literature.
Then, less important data should be discussed respectively with the literature, that is, with current and quality research articles. It is expected that there will be no review article in the discussion section.
Limitation is given and the subject is closed with conclusion.

Round 2

Reviewer 2 Report

The results of morphological analysis of neurons are not acceptable without proper quantitative analysis, so this part needs to removed from the manuscript and therefore discussion and conclusion sections also should be revised based on the revision of results

Author Response

Response to Reviewer 2 Comments

Dear Reviewer:

Thank you very much for your constructive comments concerning our manuscript entitled “The effects of intraoperative hypothermia on postoperative cognitive function in the rat hippocampus and its possible mechanism” (ID: brainsci-1498267).

Those comments are all valuable and very helpful for revising and improving our paper, as well as the important guiding significance to our researches. We have studied comments carefully and have made corrections which we hope meet with approval. The responds to the Reviewer’s comments are as follows:

Point 11: The results of morphological analysis of neurons are not acceptable without proper quantitative analysis, so this part needs to be removed from the manuscript and therefore discussion and conclusion sections also should be revised based on the revision of results.

Response 11: Thank you very much for your comment. After consulting several pathologists and experts, we have done quantitative analysis of injured neurons in the hippocampal dentate gyrus (DG) region. Injured neurons were identified by their neuronal pyknosis and necrosis with HE staining. Brain slices of four rats in each group were selected for quantitative analysis of HE staining. For each brain slice, Image-Pro Plus software was used for three counts, and its average value was taken as the value of the brain slice. Afterwards, the above results were subjected to one-way analysis of variance (ANOVA) analysis followed by Tukey’s post hoc test. We have revised the corresponding part of the results, which you could find in the manuscript. Revised portion are marked in red in the manuscript.

Special thanks to you for your valuable comments. We tried our best to improve the manuscript and made some changes in the manuscript.  These changes will not influence the content and framework of the paper. And here we did not list the changes but marked in red in revised paper. Attached please find the revised version, which we would like to submit for your kind consideration. We appreciate for Reviewer’s warm work earnestly, and hope that the correction will meet with approval. Once again, thank you very much for your comments and suggestions.

Best regards.

Yours sincerely,

Tianjia Li, MD.

Department of Anesthesiology, Peking Union Medical College Hospital,

Chinese Academy of Medical Sciences and Peking Union Medical College, Beijing, China

No. 1, Shuaifuyuan, Dongcheng District, Beijing, China

Phone: 010-69155591.

Email address: litianjia@pumch.cn

Reviewer 3 Report

Thank you for the revision. This form is acceptable.

Author Response

Response to Reviewer 3 Comments

Dear Reviewer:

Thank you very much for your positive comments concerning our manuscript entitled “The effects of intraoperative hypothermia on postoperative cognitive function in the rat hippocampus and its possible mechanism” (ID: brainsci-1498267).

We noticed that your comment was “Thank you for the revision. This form is acceptable”, however, the “Open Review” was changed from “I would like to sign my review report” to “I would not like to sign my review report”. We want to know if it is a submission error. We appreciate you very much if you could check it. Once again, thank you very much for your comments and suggestions.

Best regards.

Yours sincerely,

Tianjia Li, MD.

Department of Anesthesiology, Peking Union Medical College Hospital,

Chinese Academy of Medical Sciences and Peking Union Medical College, Beijing, China

No. 1, Shuaifuyuan, Dongcheng District, Beijing, China

Phone: 010-69155591.

Email address: litianjia@pumch.cn